# A Technical Report on the Potential Effects of Heat Stress on Antioxidant Enzymes Activities, Performance and Small Intestinal Morphology in Broiler Chickens Administered Probiotic and Ascorbic Acid during the Hot Summer Season

**DOI:** 10.3390/ani13213407

**Published:** 2023-11-02

**Authors:** Victory Osirimade Sumanu, Vinny Naidoo, Marinda Oosthuizen, Joseph Panashe Chamunorwa

**Affiliations:** 1Department of Anatomy and Physiology, Faculty of Veterinary Science, University of Pretoria, Onderstepoort, Pretoria P.O. Box 14679, South Africa; joseph.chamunorwa@up.ac.za; 2Department of Paraclinical Sciences, Faculty of Veterinary Science, University of Pretoria, Onderstepoort, Pretoria P.O. Box 14679, South Africa; vinny.naidoo@up.ac.za; 3Department of Veterinary Tropical Diseases, Faculty of Veterinary Science, University of Pretoria, Onderstepoort, Pretoria P.O. Box 14679, South Africa; marinda.oosthuizen@up.ac.za

**Keywords:** ascorbic acid, probiotic, antioxidant enzymes, performance, small intestinal morphology

## Abstract

**Simple Summary:**

Thermal stress is an environmental factor that negatively affects poultry production globally. It elicits behavioural and physiological changes in broiler chickens, hence the need to find ways of ameliorating its detrimental effects which are mainly expressed as oxidative stress. This study was designed as an intervention on the effect of heat stress during the hot summer season in broiler chickens’ production using probiotic and ascorbic acid as anti-stress agents. From the results, probiotic and/or ascorbic acid were effective in enhancing the antioxidant enzyme activities and performance of the broiler chickens. This study stands as a basis for application in animal production trials with a larger sample size.

**Abstract:**

Oxidative stress negatively affects the welfare of broiler chickens leading to poor productivity and even death. This study examined the negative effect of heat stress on antioxidant enzyme activities, small intestinal morphology and performance in broiler chickens administered probiotic and ascorbic acid during the hot summer season, under otherwise controlled conditions. The study made use of 56 broiler chickens; which were divided into control; probiotic (1 g/kg); ascorbic acid (200 mg/kg) and probiotic + ascorbic acid (1 g/kg and 200 mg/kg, respectively). All administrations were given via feed from D1 to D35 of this study. Superoxide dismutase, glutathione peroxidase and catalase activities were highly significant (*p* < 0.0001) in the treatment groups compared to the control. Performance indicators (water intake and body weight gain) were significantly higher (*p* < 0.05) in the probiotic and probiotic + ascorbic acid group. The height of duodenal, jejunal and ileal villi, and goblet cell counts of broiler chickens were significantly different in the treatment groups. In conclusion, the study showed that heat stress negatively affects the levels of endogenous antioxidant enzymes, performance and the morphology of small intestinal epithelium, while the antioxidants were efficacious in ameliorating these adverse effects.

## 1. Introduction

Thermal stress is a considerable problem experienced globally, especially during the summer in the sub-tropics and tropics [1,2]. Heat stress in poultry production severely affects the profitability of both small- and large-scale farms due to lower productivity and growth in the stressed birds [3,4]. High ambient temperature (AT) induces an increase in the production of reactive oxygen species (ROS) above the levels that the body can tolerate, with resultant oxidative stress [5]. Oxidative stress results from an imbalance between free radicals and antioxidant enzymes in the body system [6]. Oxidative stress induces DNA mutations, damages the respiratory chain and alters the permeability of the membrane in the mitochondria. Of the various effects of oxidative stress, one is lipid peroxidation where the free radicals tend to extract electrons from lipids in the cell membrane which leads to damage in all cell types [7].

Endogenous antioxidants like catalase (CAT), glutathione peroxidase (GPx) and superoxide dismutase (SOD), are important enzymes that improve the immune status of individuals [8,9]. While the concentrations of these enzymes are adequate for preserving normal cellular functionality under normal physiology; however, under increased ROS production such as during heat stress, they are often depleted necessitating the need for exogenous antioxidants [10,11] or for precursors to be administered [12,13]. The small intestine is a vital organ involved with the role of digestion and absorption of nutrients from the diet, alterations in its function negatively impact other organ systems’ functions in an individual. Oxidative stress adversely affects the intestinal integrity of broiler chickens necessitating the administration of anti-stress agents to protect the morphology of the intestine [14,15]. Although some osmolytes (betaine), phytochemicals (lycopene), electrolytes (bicarbonate ions), minerals and vitamins were reported to be effective in mitigating heat stress, not all were found to be effective in the gut [16].

Probiotics generally provide protection against gut pathogens, improve the intestinal microbial balance, and modulate the immune system [17,18]. A probiotic is a living organism that elicits an antioxidant, anti-stress and a growth-promoting effect in a host. They are rich in vitamin B, protein, fat and enzymes such as phytase and cellulase [19,20,21]. Ascorbic acid, a water-soluble vitamin, serves as both an antioxidant and anti-stress agent, it protects the body against ROS’s deleterious effects by preventing the formation of excessive free radicals above what the body can cope with [22].

Our aim was to intervene in broiler chicken production by alleviating the adverse effects of heat stress during summer under semi-controlled conditions via the administration of probiotic and/or ascorbic acid in their feed. This study is interventionist by design and allowed the animals to be closely monitored, repeatedly sampled and to minimize the potential of additional stressors associated with large group housing.

## 2. Materials and Methods

### 2.1. Experimental Animals, Management, and Environmental Conditions

A total of 56 (day-old) male and female broiler chicks (Ross 308) with an average weight of 40 g that were apparently healthy according to chick classification methods, were purchased from Gauteng poultry farmers Alfa chicks (215 Loerie Street, Haakdoornboom, Pretoria, South Africa). The sample size was calculated using G*Power software version 3.1.9.7, with an effect size f value of 0.59, critical F of 2.78 and power of 0.95.

The broiler chicks were kept under an intensive management system with prevailing natural environmental conditions at the Onderstepoort Veterinary Animal Research Unit, Faculty of Veterinary Science, University of Pretoria, South Africa. The guidelines of the Research Ethics Committee and Animal Ethics Committee of the University of Pretoria, South Africa (REC050-20) were adhered to. Electric sensors (Hobo) were used to monitor the ambient temperature of the pens. Infra-red bulbs were used as a source of warmth during brooding for 14 days at 32 °C. The relative humidity and ambient temperature of the pen were 75–80% and 30–36 °C, respectively. The broiler chicks were allowed access to feed (Epol commercial feed, Gauteng, South Africa) and water ad libitum. The broiler chicks were divided into 4 groups of 14 each and they were fed with broiler starter (D1–D14), broiler grower (D15–D25) and broiler finisher (D26–D35). The feed analysis is presented in Table 1.

The pen was made of concrete floor, cement blocks, with tunnel ventilation and an aluminium roof. Foot baths (F10SC, Health and Hygiene (Pty) Ltd., Roodepoort, South Africa), protective clothing and footwear were provided for the assistants to enhance biosecurity.

### 2.2. Experimental Design

The broiler chickens (n = 56) were divided into control; probiotic Saccharomyces cerevisiae (1 g/kg of feed) [20]; ascorbic acid (200 mg/kg of feed) [22]; probiotic (Saccharomyces cerevisiae); and ascorbic acid groups (1 g/kg of feed and 200 mg/kg of feed), respectively. These agents were mixed with the feed of the chickens during the study, taking the above dose rates into consideration. Colour markings and wing tags were used as a means of identification for each broiler chicken.

### 2.3. Measurement of Oxidative Stress Biomarkers

#### 2.3.1. Superoxide Dismutase Activity

Seven broiler chickens were randomly selected and fasted at D35 of this study. They were sacrificed via euthanasia with the use of a gas mixture (35% CO_2_, 35% N_2_ and 30% O_2_). The breast muscle tissues were severed, and 5 g of the tissue was placed in tubes and homogenised. SOD activity was determined using an ELISA kit (BIOCOM Africa, Clubview, Gauteng, South Africa). Briefly, reagent 1 (1 mL) was added to 2 mL of the samples in each well. Reagent 2 (0.1 mL), reagent 3 (0.1 mL) and reagent 4 (0.1 mL) were added to the wells. The mixture was gently mixed by shaking and incubated for 40 min at 37 °C. Chromogenic agent (2 mL) was added to each well and mixed for 10 min. The optical density (OD) was determined at 550 nm using a spectrophotometer (UV-VIS, Perlong Medical Equipment Co., Ltd., Nanjing, China).

#### 2.3.2. Catalase Activity

The activity of catalase (CAT) was determined with the aid of a CAT ELISA kit (BIOCOM Africa, Clubview, Gauteng, South Africa). Briefly, reagent 1 (200 µL) was added to about 20 µL of the samples in each well and incubated for 5 min at 37 °C. Reagent 2 (20 µL) was added to the wells and mixed for 1 min; 200 µL of reagent 3 application solution and reagent 4 (20 µL) were added, respectively, to each well. It was mixed and incubated for 10 min (at room temperature). OD was measured at 405 nm with a spectrophotometer.

#### 2.3.3. Glutathione Peroxidase Activity

The activity of glutathione peroxidase (GPx) was determined with the aid of a GPx ELISA kit (BIOCOM Africa). Briefly, 50 µL of the sample was added to the wells and biotinylated detection antibodies (50 µL) were added to each well immediately. The mixture was incubated for 45 min at 37 °C; after that, it was aspirated and washed thrice. HRP conjugate (100 µL) was added to the wells and incubated (for 30 min at 37 °C). The mixture was washed and 90 µL of substrate reagent was added. Stop solution (50 µL) was also added to each well and the OD was determined at 450 nm immediately after, with the aid of a spectrophotometer.

### 2.4. Measurement of Performance Parameters

#### 2.4.1. Measurement of Feed Intake

Broiler chickens’ feed intake was measured daily at 07:00 h. The feed weight before placement, and the remnant feed after 24 h of feeding were measured using a Digital Precision (Mettler Toledo^®^) weighing balance (Greifensee, Switzerland). Absolute feed intake for each day was calculated as the difference between the amount of left-over feed and the amount of feed supplied to the broiler chickens.

#### 2.4.2. Measurement of Water Intake

A graduated cylinder (Rutland Industries, 8 Theodore Road, Benrose, Johannesburg, South Africa) was used to measure the quantity of water before placement and after consumption (24 h later).

#### 2.4.3. Measurement of Body Weight Gain

Upon arrival at the poultry pen, each broiler chicken was weighed at D1 and these values served as the initial body weight. The broiler chickens’ weights were measured once weekly to determine the average weekly body weight gain. The study was terminated after the broiler chickens were euthanized via the use of a gas mixture (35% CO_2_, 35% N_2_ and 30% O_2_).

### 2.5. Measurement of Small Intestinal Morphology

At D35 of the experiment, seven broiler chickens were selected at random from various groups and fasted (for 12 h). About 3 cm length of the middle portion of the ileum and jejunum and the descending portion of the duodenum for each broiler chicken were taken and stored in a tube containing 10% solution of buffered neutral formaldehyde (pH 7.2–7.4). After dehydration, the samples were placed in paraffin blocks, they were sectioned and stained with periodic acid–Schiff reagent according to the method of Luna [23]. The crypt width and height, villus width and height, villus and crypt surface areas and goblet cells of the small intestinal epithelium were measured at 100× magnification (Sigma Scan, Jandel Scientific, San Rafael, CA, USA).

Transmission electron microscopy was used to evaluate the morphology of the goblet cells. 0.5 × 0.5 × 0.5 cm tissue blocks of the ileum, jejunum and duodenum were taken and inserted into a tube containing 2.5% glutaraldehyde and processed according to the standard method of Cheville and Stasko [24]. Briefly, tiny pieces of the small intestinal tissue were placed into glutaraldehyde and fixed at room temperature for 1 h, and then were post fixed in osmium tetroxide. The fixed cells were embedded in agar and afterward were dehydrated in ethanol-graded concentrations (with propylene oxide). A glass knife was used to cut the tissue sections from blocks for thinning. The thin sections were placed on copper grids; uranyl acetate and lead citrate were used for impregnation and scoping. Stained sections were visualised using a JEOL 1400 electron microscope operated at 80 kV.

### 2.6. Data Analyses

Antioxidant enzymes and morphological data were subjected to one-way analysis of variance (ANOVA), while the performance parameters were subjected to two-way ANOVA followed by Tukey’s multiple comparison post hoc test to compare differences between the treatment and control groups’ means. The data obtained were expressed as mean ± standard error of the mean (Mean ± SEM). Version 27 (Armonk, NY, USA: IBM Corp.) SPSS Statistics for Windows software was used for the analysis. Values of *p* ˂ 0.05 were considered significant.

## 3. Results

### 3.1. Superoxide Dismutase Enzyme Activity

There were no differences (*p* > 0.05) in SOD activity obtained between the probiotic, probiotic + ascorbic acid and ascorbic acid groups. The activity of SOD was higher (*p* < 0.0001) in the probiotic, probiotic + ascorbic acid and ascorbic acid groups of broiler chickens when compared to the control group (Figure 1).

### 3.2. Catalase Enzyme Activity

There was a significant difference (*p* < 0.05) in CAT activity obtained between the probiotic and ascorbic acid groups. The activity of CAT was higher (*p* < 0.0001) in the probiotic, probiotic + ascorbic acid and ascorbic acid groups of broiler chickens in comparison with the control group (Figure 2).

### 3.3. Glutathione Peroxidase Enzyme Activity

The activity of GPx was higher (*p* < 0.0001) in the probiotic + ascorbic acid, probiotic and ascorbic acid groups of broiler chickens when compared to the control. There was no difference (*p* > 0.05) in GPx activity obtained between the treatment groups (Figure 3).

### 3.4. Feed and Water Intake, and Body Weight Gain

Values of feed intake recorded in the treatment groups were not different (*p* > 0.05) when compared with the control group. Water intake recorded in the ascorbic acid and probiotic + ascorbic acid groups was higher (*p* < 0.05) when compared with the values obtained in the control group at D 21 to D 35. Body weight gain of broiler chickens in the probiotic + ascorbic acid and ascorbic acid groups were not significantly different in comparison with the control, while that of the probiotic group was higher (*p* < 0.05) when compared with the control group at D 28 to D 35 (Table 2).

### 3.5. Morphological Analysis

Morphological analysis of the villus width and height revealed that villus width and height were different (*p* < 0.001) in the probiotic and probiotic + ascorbic acid groups in comparison with the control group. The crypt width and depth of the ileum, jejunum and duodenum were greater (*p* < 0.0001) in the probiotic and probiotic + ascorbic acid groups in comparison with the control. The probiotic and probiotic + ascorbic acid groups had significantly different surface areas of villi and crypts of the small intestine when compared to the control group. The number of goblet cells of the duodenum, ileum and jejunum was greater (*p* < 0.001) in the treatment groups in comparison with the control group during this study (Table 3).

The histological representations of the effect of heat stress on the jejunal epithelium of broiler chickens in the treatment and control groups are shown in Figure 4 and Figure 5.

## 4. Discussion

### 4.1. Antioxidant Enzyme Activities

The decreased values of SOD, CAT and GPx were evident that the broiler chickens were negatively affected by heat stress. Increased ROS production which occurs during heat stress generally impacts the activities of endogenous enzymes if no exogenous source is used for augmentation in broiler chickens [25]. As shown from previous studies, heat stress stimulates the process of lipid peroxidation which occurs due to the depletion of the antioxidant defence system in broiler chickens [26]. Xue et al. [27] also reported a drastic depletion in the antioxidant activities of broiler chickens exposed to heat stress. They attributed this to the fact that depletion of the antioxidant defence system occurs during heat stress, resulting in cellular damage from rapid oxidation processes. From the treatment groups, the probiotic group had the highest SOD, CAT and GPx activities followed by the combination and the ascorbic acid groups. Antioxidants directly react with free radicals converting them to nonradical products that are stable [28].

According to Winiarska-Mieczan et al. [29], exogenous antioxidants induce the activities of SOD, CAT and GSH in the muscle tissue of broiler chickens leading to the protection of the cellular membranes from peroxidation. Kumar et al. [30] stated in their findings that the use of ascorbic acid and Moringa oleifera in broiler chickens raised in the tropics augmented the activities of CAT, SOD and GSH, and lipid peroxidation was reduced significantly. This was based on the ability of the above antioxidants to attach to the cytoplasmic membrane and inhibit oxidases. Deng et al. [31] reported that the supplementation of broiler chickens’ diet with yeast probiotic improved their antioxidant status, which is suggestive of their strong antioxidant potential as compared to selenium which had no effect. This was due to the ability of yeast probiotic to scavenge free radicals during normal metabolism in the body.

### 4.2. Performance Indicators

Generally, it is assumed that the non-significant value in feed intake observed between the control and treatment groups might be due to the degree of high ambient temperature the broiler chickens were exposed to as water intake is prioritized over feed intake during prolonged exposure to thermal stress. The finding agrees with that of Moataz et al. [32], who reported decreased feed intake in layers exposed to heat stress when compared with the groups that were administered the *Bacillus subtilis* probiotic. He associated the decrease to the degree of heat stress the layers were exposed to.

The decreased water intake observed in the control may be based on their low feed intake, as the quantity of feed intake may be directly proportional to water intake to further enhance digestibility. Ascorbic acid-administered group had an increase in water intake, which could be supported by the fact that feed intake stimulates an increase in water intake. Generally, increased water intake obtained in the treatment groups could also be assumed as a thermoregulatory measure to rid the body of excessive heat during the period of study. The increase in water intake recorded in the treatment groups agrees with that of Egbuniwe et al. [33] who reported an increase in water intake in the group of chickens administered ascorbic acid and/or betaine than the control group. They also attributed the increase to a means of thermoregulation.

Body weight gain obtained in the control group was lowest during the study period. This may be in line with the lower feed (though not significantly different) and water intake obtained when compared with the probiotic and probiotic + ascorbic acid groups. It is important to note that the Ross breed of broiler chickens utilized for this study was reported to attain the highest weight of 2.5 kg at D42 under a thermoneutral condition according to the Ross broiler pocket guide [34], but the probiotic-supplemented group had the highest body weight gain (2.8 kg) at D35 (5 weeks) of this study. This increase may be linked to the fact that probiotics are gut effective and increased water intake may further enhance the rate at which ingested feed is utilized for optimum body weight gain Aluwong et al. [18]. Additionally, probiotics could be vital in the process of nutrient digestion due to the optimization of the gut microflora, although its effect on the transportation of nutrients within the body system remains a grey area for further examination.

To the best of our knowledge, most researchers conducted their studies using probiotics to optimize the rate of performance in broiler chickens within 42 days (6 weeks), but our study achieved better outcomes within 35 days with an average weight of 2.7 kg being recorded in the probiotic-administered group. Olnood et al. [35] and Aluwong et al. [36] speculated that the administration of probiotics via gavage is the fastest route that promotes broiler chickens’ performance. Nevertheless, we administered the yeast probiotic via feed (oral route) and from the results, improved performance indices were observed in this group of broiler chickens during our study. Wang et al. [37] reported that probiotics (*Bacillus subtilis*) increased the body weight gain of broiler chickens to 2.3 kg at D43 of their study and this was attributed to the gut enhancement effect of the additive. Increased body weight gain attained in the ascorbic acid group might be due to the improved feed and water intake recorded in this group based on its antioxidant effects in improving wellness which could further stimulate an improvement in performance. The finding is in line with that of Egbuniwe et al. [33] who reported an increase in body weight gain in a group of chickens administered ascorbic acid singly during the heat stress, they presumed the increase to be a result of the anti-stress effect of the agent.

### 4.3. Small Intestinal Morphology

In addition to the changes in the specific markers, the villus height and width, crypt depth and width were shortened, while the villus and crypt surface areas and goblet cell count of the duodenum, jejunum and ileum in the control group were decreased. Intestinal distortion is basically a primary response to heat stress, and this could further compromise the barrier function and integrity of the intestine in broiler chickens as observed in this group in comparison with the treatment group.

Also, the jejunal epithelia lacteal of the control group of broiler chickens was distorted based on the adverse effects of heat stress. The epithelia lacteal is responsible for the absorption of fatty acids and cholesterol in the small intestine, therefore, any form of alteration in their morphology could affect the process of absorption. The gastrointestinal tract has been reported to be more susceptible to any form of stressors, especially heat stress because it impairs the integrity of the epithelium via decreasing crypt depth and villus height [38,39,40]. Zampiga et al. [41] stated that heat stress induces damage in the small intestinal mucosa of broiler chickens by increasing the permeability of the intestine to endotoxins which subsequently leads to a reduction in their growth performance.

The treatment groups, especially the probiotic-administered broiler chickens had an improved morphology in comparison with the ascorbic acid group. It could be speculated that ascorbic acid was not as effective as probiotics in the gut due to the poor epithelial turnover obtained during the study. Enhanced epithelial turnover directly influences the height of the small intestinal epithelium via the stem cells of the crypts of Lieberkuhn that is responsible for the rapid renewal of the functional villus epithelium as yeast probiotics generally improve the intestinal epithelial cell integrity in broiler chickens [40,41,42]. In broiler chickens, the stimulation of mixing movements exposes digesta to the enzymes and improves the process of digestion and absorption of nutrients, this could further enhance the growth performance of broiler chickens. The presence of adequate mucins in the goblet cells of broiler chickens treated with probiotics could have influenced their growth rate as they are responsible for protecting the epithelium and regulating the concentration and passage of some immune mediators (antimicrobial peptides), ions and water [43]. Sen et al. [44] suggested that increased intestinal morphology enhances the digestion and absorption capacity of the small intestine in broiler chickens supplemented with *Bacillus subtilis*.

Even though changes to morphology can account for part of the mechanisms of action of probiotics, the specific mechanisms by which it influences transport across the epithelium remain an area for further research.

## 5. Conclusions

Heat stress negatively affects antioxidant enzymes, performance and small intestinal morphology of the broiler chickens, although probiotic and/or ascorbic acid administrations were efficacious in alleviating the detrimental effects of heat stress during this study. This was evident by an increase in endogenous antioxidant enzyme activities. The probiotic-administered group had the highest body weight gain, followed by the co-administered and ascorbic acid groups, respectively. The morphology of the small intestinal epithelium was enhanced by these treatments in the phase of heat stress exposure. Also, it would be important to determine if the effect of probiotic and/or ascorbic acid evident under controlled conditions would be translatable to open ventilated broiler houses where birds would be exposed to both production conditions and higher thermal conditions.

## Figures and Tables

**Figure 1 animals-13-03407-f001:**
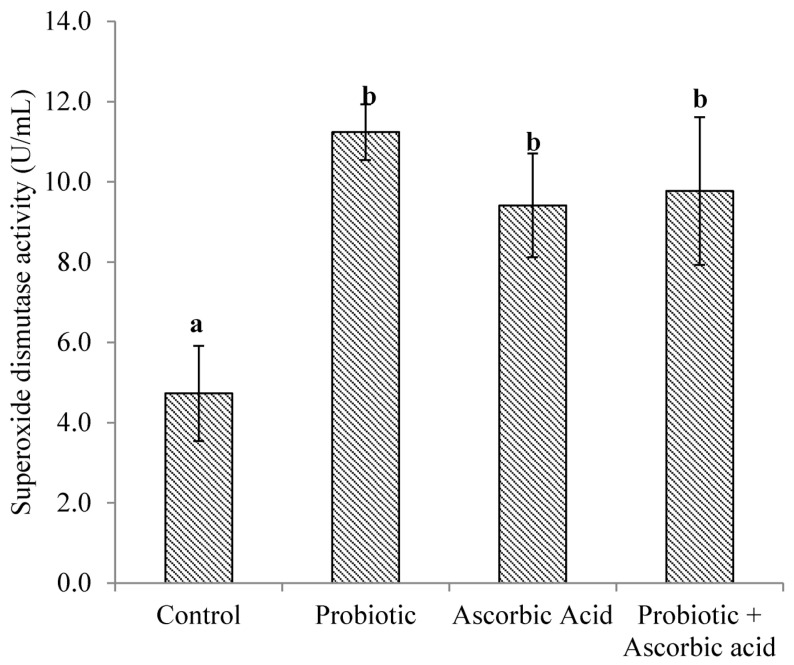
Superoxide dismutase activity in broiler chickens’ muscle tissue exposed to heat stress and administered probiotic and ascorbic acid. *p* < 0.001. Data are expressed as Mean ± SEM. Vertical bars with different superscript letters ^a,b^ are significantly different at *p* < 0.05.

**Figure 2 animals-13-03407-f002:**
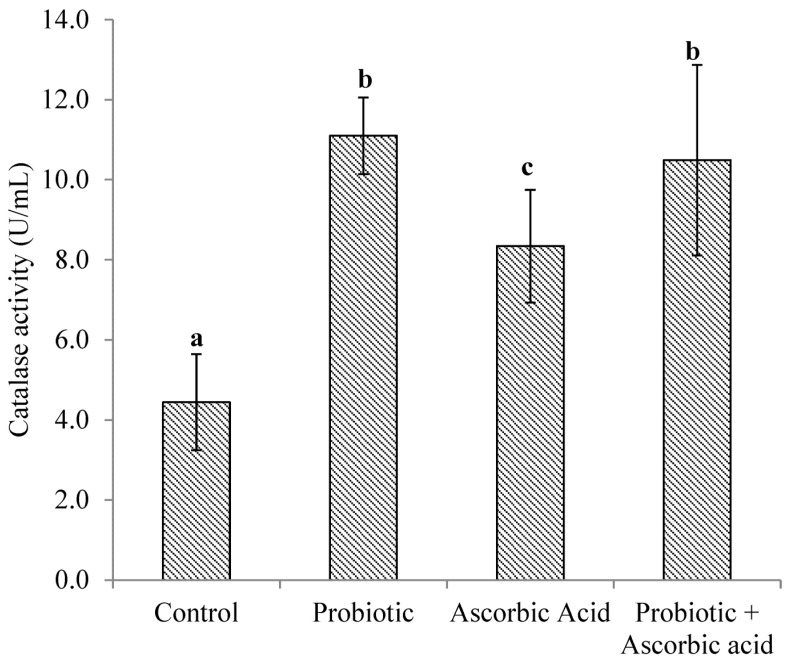
Catalase activity (U/mL) in broiler chickens muscle tissue exposed to heat stress and administered probiotic and ascorbic acid. *p* < 0.01. Data are expressed as Mean ± SEM. Vertical bars with different superscript letters ^a,b,c^ are significantly different at *p* < 0.05.

**Figure 3 animals-13-03407-f003:**
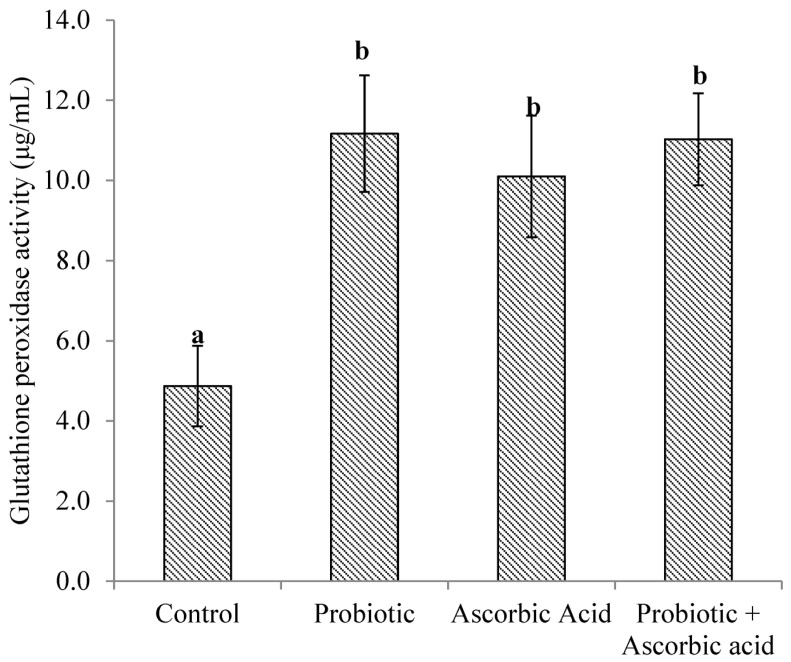
Glutathione peroxidase activity in broiler chickens’ muscle tissue exposed to heat stress and administered probiotic and ascorbic acid. *p* < 0.002. Data are expressed as Mean ± SEM. Vertical bars with different superscript letters ^a,b^ are significantly different at *p* < 0.05.

**Figure 4 animals-13-03407-f004:**
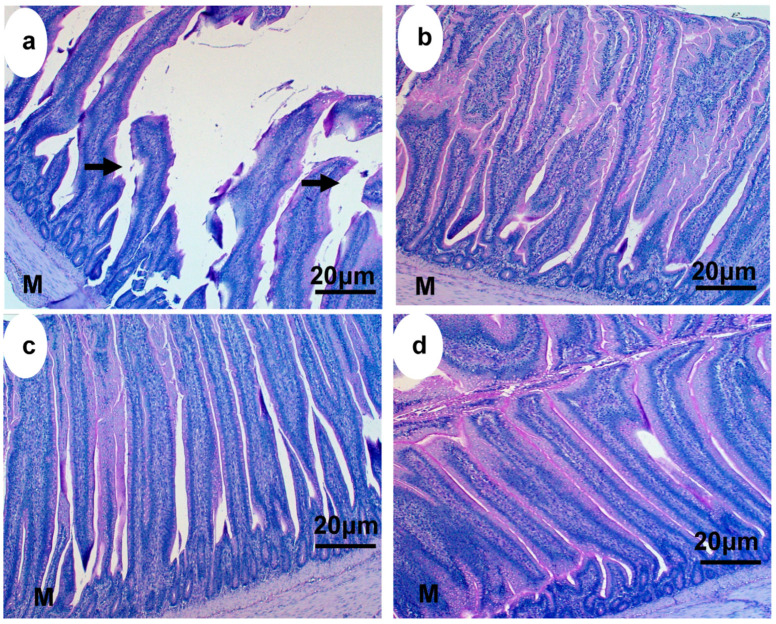
Light microscopy showing the jejunal epithelia of broiler chickens exposed to heat stress: (**a**) control group, (**b**) probiotic, (**c**) ascorbic acid, and (**d**) probiotic and ascorbic acid. Arrows = widening of capillary lacteal due to distortion of epithelia, M = muscularis mucosa.

**Figure 5 animals-13-03407-f005:**
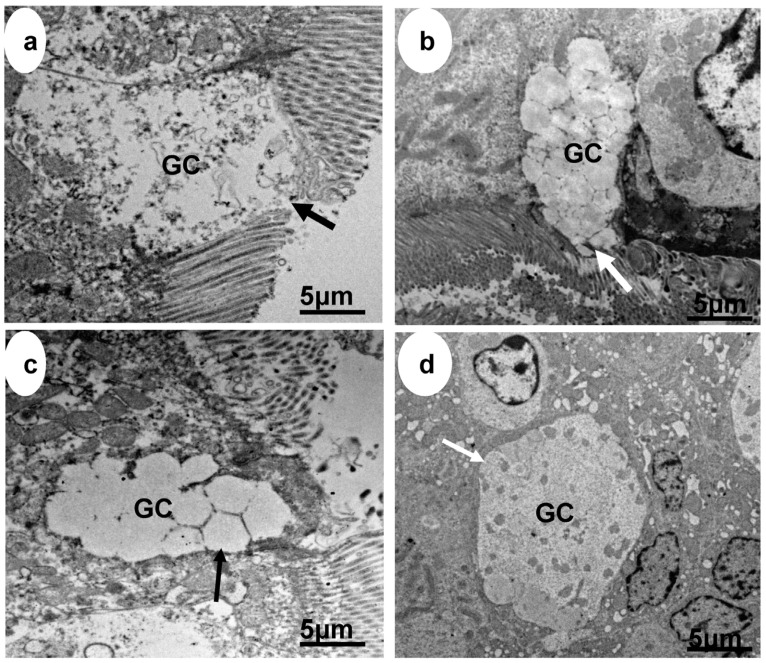
Electron micrographs of jejunum of broiler chickens exposed to heat stress (**a**) without treatment, a distorted goblet cell without a defined contour and absence of mucin (thick black arrow); (**b**) treated with probiotic, showing the goblet cell and presence of mucin (thick white arrow); (**c**) treated with ascorbic acid, and there was no presence of mucin but presence of dilations within the lobules of the goblet cell (thin black arrow); and (**d**) treated with probiotic and ascorbic acid, in which the goblet cell has a well-defined contour (thin white arrow), GC = goblet cell.

**Table 1 animals-13-03407-t001:** Proximate composition of feed.

Feed Composition	Starter	Grower	Finisher
Proximate analysis (%)			
Protein	18.22	18.23	18.24
Total lysine	1.67	1.68	1.69
Total methionine	0.63	0.64	0.65
Fat	3.45	3.46	3.47
Fibre	5.54	5.55	5.56
Calcium	0.58	0.59	0.6
Phosphorus	0.52	0.53	0.54
Metabolizable energy (kcal/kg)	2910	2980	3000

**Table 2 animals-13-03407-t002:** Performance indicators in broiler chickens treated with probiotic and ascorbic acid.

	Group	Day 0–7	Day 8–14	Day 15–21	Day 22–28	Day 29–35
Feed intake (g)	Control	675.14 ± 157.20	1027.57 ± 58.95	1836.71 ± 80.19	2505.43 ± 70.01	2656.86 ± 45.88
	Probiotic	674.43 ± 137.40	981.57 ± 64.21	1923.00 ± 134.98	2821.86 ± 41.10	3126.71 ± 56.54
	Ascorbic acid	708.71 ± 153.98	1045.00 ± 64.21	2327.71 ± 175.52	2626.00 ± 31.61	2792.29 ± 70.02
	Prob + AA	672.29 ± 139.07	1322.29 ± 71.42	2066.57 ± 87.43	2571.43 ± 16.12	2920.71 ± 62.78
Water intake (mL)	Control	1400.00 ± 293.58^a^	2542.86 ± 184.98^a^	4585.71 ± 280.67^a^	5928.57 ± 184.80^a^	7350.00 ± 227.26^a^
	Probiotic	2000.00 ± 243.98^b^	2971.43 ± 176.90^a^	4900.00 ± 325.87^a^	6828.57 ± 164.34^b^	8628.57 ± 316.77^a^
	Ascorbic acid	1942.86 ± 220.23^a^	3200.00 ± 211.57^b^	5142.86 ± 348.37^b^	7057.14 ± 165.99^b^	8771.43 ± 222.23^b^
	Prob + AA	1928.57 ± 229.61^a^	2971.43 ± 178.24^a^	5085.71 ± 317.30^b^	6871.43 ± 171.43^b^	7471.43 ± 153.86^b^
Body weight gain (g)	Control	181.50 ± 2.24	505.86 ± 15.14	1031.21 ± 29.89	1756.50 ± 48.23^a^	2138.50 ± 68.02^a^
	Probiotic	183.93 ± 1.16	515.57 ± 19.33	1289.07 ± 90.23	1822.41 ± 41.00^b^	2730.79 ± 55.26^b^
	Ascorbic acid	190.07 ± 1.95	521.14 ± 16.33	1063.14 ± 30.99	1768.86 ± 45.56^a^	2321.71 ± 58.36^a^
	Prob + AA	185.07 ± 2.93	522.21 ± 15.89	1077.64 ± 34.02	1629.02 ± 54.73^a^	2432.64 ± 58.82^b^

Mean ± SEM with different superscript letters ^a,b^ within the same column are significantly different at *p* < 0.01. Prob + AA = Probiotic + Ascorbic acid, n = 14.

**Table 3 animals-13-03407-t003:** Histomorphometry of the duodenum, ileum and jejunum of broiler chickens treated with probiotic and/or ascorbic acid during heat stress.

	Parameters	Control	Probiotic	Ascorbic Acid	Probiotic + AA
Duodenum	Villus height (μm)	652.18 ± 39.94^c^	1706.92 ± 129.15^a^	1427.69 ± 66.69^b^	1230.55 ± 81.85^b^
	Villus width (μm)	117.33 ± 6.98^b^	288.21 ± 53.41^a^	173.55 ± 8.23^b^	278.15 ± 44.77^a^
	Crypt depth (μm)	95.61 ± 12.29^c^	329.45 ± 37.26^a^	267.20 ± 53.10^b^	324.23 ± 43.89^b^
	Crypt width (μm)	48.16 ± 4.86^d^	159.57 ± 16.87^a^	125.07 ± 20.47^c^	140.95 ± 25.55^b^
	Villus height/crypt depth	2.40 ± 0.10^c^	4.03 ± 0.62^a^	3.24 ± 1.24^b^	3.03 ± 0.66^b^
	Villus SA (mm^2^)	5.40 ± 1.29^c^	16.00 ± 2.98^a^	7.50 ± 3.21^b^	10.80 ± 1.32^b^
	Crypt SA (mm^2^)	2.80 ± 0.66^c^	12.80 ± 2.13^a^	3.80 ± 1.24^c^	8.40 ± 1.03^b^
	Goblet cells	58.00 ± 13.99^c^	101.00 ± 4.92^b^	108.20 ± 9.32^a^	102.80 ± 12.14^b^
Ileum	Villus height (μm)	284.58 ± 56.72^d^	1025.07 ± 66.80^a^	678.08 ± 58.14^c^	865.22 ± 68.37^b^
	Villus width (μm)	84.70 ± 19.19^c^	236.10 ± 41.28^a^	107.93 ± 8.03^b^	230.98 ± 19.38^a^
	Crypt depth (μm)	98.95 ± 5.09^d^	159.73 ± 25.54^b^	135.13 ± 21.12^c^	231.73 ± 35.60^a^
	Crypt width (μm)	40.80 ± 5.94^c^	88.70 ± 10.96^a^	65.38 ± 5.05^b^	92.28 ± 7.71^a^
	Villus height/crypt depth	1.22 ± 0.01^c^	2.41 ± 0.10^a^	1.49 ± 5.01^b^	2.00 ± 1.01^b^
	Villus SA (mm^2^)	3.60 ± 0.93^b^	10.40 ± 1.44^a^	7.60 ± 1.60^a^	9.00 ± 1.22^a^
	Crypt SA (mm^2^)	3.60 ± 0.68^b^	7.40 ± 1.12^a^	3.60 ± 1.21^b^	7.40 ± 0.93^a^
	Goblet cells	52.40 ± 9.69^c^	183.20 ± 20.61^a^	125.40 ± 22.65^b^	122.20 ± 13.40^b^
Jejunum	Villus height (μm)	362.95 ± 68.90^c^	1205.70 ± 105.67^a^	1010.33 ± 62.26^b^	1066.50 ± 51.31^b^
	Villus width (μm)	102.81 ± 11.09^c^	204.58 ± 20.54^b^	202.58 ± 37.60^b^	220.48 ± 35.19^a^
	Crypt depth (μm)	123.46 ± 11.16^c^	303.43 ± 36.83^a^	240.39 ± 39.34^b^	279.19 ± 26.99^b^
	Crypt width (μm)	48.47 ± 4.33^c^	98.71 ± 14.12^a^	84.21 ± 5.62^b^	88.41 ± 9.57^b^
	Villus height/crypt depth	1.10 ± 0.01^c^	2.44 ± 3.00^a^	1.90 ±1.02^b^	2.00 ± 0.01^b^
	Villus SA (mm^2^)	2.70 ± 0.49^c^	10.20 ± 1.39^b^	11.40 ± 3.06^a^	9.80 ± 0.86^b^
	Crypt SA (mm^2^)	2.80 ± 0.66^c^	8.20 ± 1.28^a^	6.40 ± 1.57^b^	7.80 ± 1.02^a^
	Goblet cells	56.00 ± 8.15^c^	150.60 ± 20.92^a^	93.00 ± 9.95^b^	102.60 ± 6.37^b^

Mean ± SEM with different superscript letters ^a,b,c,d^ along the same row are significantly different. n = 7. AA = ascorbic acid, SA = surface area.

## Data Availability

Data are contained within the article.

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
