# Peer review of "A Technical Report on the Potential Effects of Heat Stress on Antioxidant Enzymes Activities, Performance and Small Intestinal Morphology in Broiler Chickens Administered Probiotic and Ascorbic Acid during the Hot Summer Season"

_animals, 2023, doi:10.3390/ani13213407_

Round 1

Reviewer 1 Report

Comments and Suggestions for Authors

The research discussed in the manuscript delves into the crucial realm of poultry production amidst the challenges posed by climate change.  The manuscript adheres to the standard structure of a scientific paper and is well-written, with clear objectives and a concise discussion of the findings. Nonetheless, I have provided detailed remarks and recommendations, which you can access in the attached PDF file.

Comments on the Quality of English Language

Need to improve, check the attached pdf file.

Author Response

The research discussed in the manuscript delves into the crucial realm of poultry production amidst the challenges posed by climate change.  The manuscript adheres to the standard structure of a scientific paper and is well-written, with clear objectives and a concise discussion of the findings. Nonetheless, I have provided detailed remarks and recommendations, which you can access in the attached PDF file.

RESPONSE: All comments are highly appreciated. The changes made are highlighted in red. Thanks a great deal.

Reviewer 2 Report

Comments and Suggestions for Authors

1. Please check the format and change it for the one used for the journal of animals (top, bottom and side margins)

2. In the abstract, start with a general sentence, then add objectives, then explain the experimental design, then explain the conclusions....

3. In table 1, change tittle to "proximate composition of feed"

4. What is the energy level?????

5. How many birds were placed per pen and how many pens????

6. Please organize your material and methods. You can look up another paper and learn from it!!!

7. Place P-values in each graph

8. I don't understand this: There was no difference (P > 0.05) in CAT activity obtained between the treatment 202 groups. The activity of CAT was higher (P < 0.0001) in the probiotic, probiotic + ascorbic 203 acid and ascorbic acid groups of broiler chickens when compared to the control group 204 (Figure 2).

9. Please organize all the images, graphs, and tables.

Comments on the Quality of English Language

Quality is okay

Author Response

  1. Please check the format and change it for the one used for the journal of animals (top, bottom and side margins).

RESPONSE: Noted. It has been done.

  1. In the abstract, start with a general sentence, then add objectives, then explain the experimental design, then explain the conclusions....

RESPONSE: Noted with thanks. The abstract has them in place. Lines 26-40.

  1. In table 1, change tittle to "proximate composition of feed"

RESPONSE: Noted. The change has been made. Line 108.

  1. What is the energy level?????

RESPONSE: 3000 kcal/kg.

  1. How many birds were placed per pen and how many pens????

RESPONSE: 14 birds were placed per pen. 4 pens were used for the research.

  1. Please organize your material and methods. You can look up another paper and learn from it!!!

RESPONSE: Noted with thanks. The author’s guideline for the journal was adhered to and some sample articles from the journal served as a guide.

  1. Place P-values in each graph

RESPONSE: Noted with thanks. The P-values have been placed in the legend of the graphs. Lines, 186, 199, 209, respectively.

  1. I don't understand this: There was no difference (P > 0.05) in CAT activity obtained between the treatment 202 groups. The activity of CAT was higher (P < 0.0001) in the probiotic, probiotic + ascorbic 203 acid and ascorbic acid groups of broiler chickens when compared to the control group 204 (Figure 2).

RESPONSE: It has been edited to read ‘There was a significant difference (P < 0.05) in CAT activity obtained between the probiotic and ascorbic acid groups. The activity of CAT was higher (P < 0.0001) in the probiotic, probiotic + ascorbic acid and ascorbic acid groups of broiler chickens in comparison with the control group (Figure 2)’.

  1. Please organize all the images, graphs, and tables.

RESPONSE: Noted with thanks. They have been edited.

Reviewer 3 Report

Comments and Suggestions for Authors

Review: Potential effects of heat stress on antioxidant enzymes activities, performance and small intestinal morphology in broiler chickens administered probiotic and ascorbic acid during the hot summer season.

 Overall Comment

The study seems to exhibit signs of pseudo replication. This term is used when there's an improper application of statistical methods due to the misidentification and use of non-independent replicates in experimental setups. The study appears to treat the 14 hens as separate observational units for statistical analysis when they might not be genuinely independent. My inference from line 97, which describes the pen, suggests that these hens were housed together, thus making their data points non-independent. Instead of considering each hen as a unique replicate, the study should perhaps treat each pen as a single replicate. The presence of pseudo replication raises concerns about the statistical soundness of the study. The resultant conclusions might not accurately reflect patterns within a broader context. The statistical significance displayed might be misleading, given the potentially inflated replicates, creating undue confidence in the outcomes. As such, the study's findings and subsequent conclusions could be deemed unreliable or at least debatable. If decisions are based on these outcomes, they might be misguided. It is imperative to address these methodological shortcomings, and a reconsideration or repetition of the experiment might be in order, ensuring the results are trustworthy and robust.

Refeences:

Hurlbert, S. H. Pseudoreplication and the design of ecological field experiments. Ecol. Monogr. 54, 187–211 (1984).

Comments on the Quality of English Language

The authors are encouraged to write concisely.

Author Response

Overall Comment

The study seems to exhibit signs of pseudo replication. This term is used when there's an improper application of statistical methods due to the misidentification and use of non-independent replicates in experimental setups. The study appears to treat the 14 hens as separate observational units for statistical analysis when they might not be genuinely independent. My inference from line 97, which describes the pen, suggests that these hens were housed together, thus making their data points non-independent. Instead of considering each hen as a unique replicate, the study should perhaps treat each pen as a single replicate. The presence of pseudo replication raises concerns about the statistical soundness of the study. The resultant conclusions might not accurately reflect patterns within a broader context. The statistical significance displayed might be misleading, given the potentially inflated replicates, creating undue confidence in the outcomes. As such, the study's findings and subsequent conclusions could be deemed unreliable or at least debatable. If decisions are based on these outcomes, they might be misguided. It is imperative to address these methodological shortcomings, and a reconsideration or repetition of the experiment might be in order, ensuring the results are trustworthy and robust.

RESPONSE: Thanks for your comments. But the study was not replicated. The hallmark of the research was to evaluate the effects of two agents which possess both anti-stress, antioxidant and/or a growth promoting effects in broiler chickens exposed to heat stress. These served as the basis for grouping the chickens into four different groups as follows; Control (without the treatment); Probiotic-treated group; Ascorbic acid treated group and lastly the combination of probiotic and ascorbic acid. See Aluwong et al., 2017, Ogbuagu et al., 2018, Sumanu et al., 2023 for similar grouping method.

This was done to arrive at a conclusion, if the administration of the agents singly or in combination would be effective in mitigating the detrimental effect of heat stress in broiler chickens.

Please, kindly see below for the biostatistician calculation that was used to arrive at the sample size.

Round 2

Reviewer 2 Report

Comments and Suggestions for Authors

1. Please add the energy levels of the diets in the table showing the feed composition

2. Why you didn't have replicates of the treatments??? How were you able to run statistics for growth performance if you just had 1 replicate pen per treatment?????

Comments on the Quality of English Language

The quality of English seems to be okay.

Author Response

  1. Please add the energy levels of the diets in the table showing the feed composition

RESPONSE: Noted with thanks. It has been included in Table 1.

  1. Why you didn't have replicates of the treatments??? How were you able to run statistics for growth performance if you just had 1 replicate pen per treatment?????

RESPONSE: Noted with thanks. Two-way ANOVA has been used to rerun the performance parameters.
